# Plastic Bronchitis—A Serious Rare Complication Affecting Children Only after Fontan Procedure?

**DOI:** 10.3390/jcm11010044

**Published:** 2021-12-23

**Authors:** Ilona Pałyga-Bysiecka, Aneta Maria Polewczyk, Maciej Polewczyk, Elżbieta Kołodziej, Henryk Mazurek, Andrzej Pogorzelski

**Affiliations:** 1First Department of Pediatrics, Swietokrzyskie Pediatric Center, 25-736 Kielce, Poland; an.tuchalska@gmail.com (A.M.P.); elzbieta.kolodziej@wszzkielce.pl (E.K.); 2Collegium Medicum, Jan Kochanowski University, 25-736 Kielce, Poland; Maciek.polewczyk@gmail.com; 3Department of Pneumology and Cystic Fibrosis, Institute of Tuberculosis and Lung Diseases, 03-700 Rabka-Zdrój, Poland; hmazurek@igrabka.edu.pl (H.M.); apogorzelski@igrabka.edu.pl (A.P.)

**Keywords:** plastic bronchitis, bronchial casts, congenital heart disease, Fontan procedure

## Abstract

Background: Plastic bronchitis (PB) may occur not only in children following palliative Fontan procedure but also in those without underlying heart disease. We aim to assess the clinical course, therapeutic measures, outcome, and follow-up of PB in children with congenital heart disease (CHD) and children without cardiac problems. Methods: This retrospective case series assessed children with PB admitted to hospital between 2015 and 2019. Parents or guardians of patients were contacted by e-mail or telephone between September 2017 and June 2019 to enquiry about recurrence of PB and strategy of treatment. The diagnosis of PB was based on the expectoration (spontaneous or during bronchoscopy) of endobronchial plugs. Results: This study delineated the clinical, histological, and laboratory features of plastic bronchitis in children following Fontan procedure (Group A) and in those without heart defects (Group B, non-CHD children). The main symptoms were cough accompanied by dyspnea, and hypoxemia with a decrease in oxygen saturation, often leading to acute respiratory failure. In children with CHD, the first episode of PB occurred at a relatively young age. Although chronic, i.e., lasting more than 3 weeks, inhaled therapy was implemented in both groups of patients, the recurrences of PB were observed. The mean time to PB recurrence after the first episode in Group A was longer than that in Group B (1.47 vs. 0.265 years, *p* = 0.2035). There was no re-episode with recurrence of PB in 3 cases out of 10 in total in Group A (30%) and 1 case out of 4 in total in Group B (25%). While the majority of children in Group A usually developed bronchial casts on the right side, the patients in Group B (without CHD) suffered from bronchial casts located only on the left side. Conclusions: Despite many similarities, clinical, histological, and laboratory studies in the children with plastic bronchitis after Fontan’s surgery and in children without heart defects suggest that there are differences in the course of the disease in patients without CHD, such as a more advanced age of the first episode of PB, the location of plastic casts on the left side, and a stronger role of inflammatory factors and mechanisms. Further research is needed to understand the pathophysiology of PB and choose the most appropriate therapy.

## 1. Introduction

Plastic bronchitis (PB) is a rare cause of a partial or complete airway obstruction, in which three-dimensional fibrinous and/or mucous casts develop in the bronchial tree. The condition has been reported at every age; however, it has been predominantly seen in children who have undergone Fontan palliative surgical procedure for congenital heart disease (CHD) [1]. PB is a serious and life-threatening complication, as is a protein losing enteropathy (PLE) [2], since both conditions are associated with a lymphatic system dysfunction [3]. Rarely has PB occurred in patients with a chronic pulmonary disease such as asthma, cystic fibrosis, lymphangiomatosis, acute chest syndrome associated with sickle cell disease, and after heart transplantation [1,4,5,6]. Several infectious agents have been shown to be involved in PB development, including human bocavirus, H1N1 influenza virus, adenovirus, SARS-CoV-2 virus, *Mycoplasma pneumoniae*, and opportunistic fungi [7,8,9,10,11,12,13,14,15,16,17]. Antibiotics recommended for PB caused by infectious bacteria include macrolides, in particular, azithromycin, which has anti-inflammatory and immunomodulatory activity [18,19].

Plastic bronchitis may initially progress in an unobtrusive manner because the symptoms are nonspecific. Patients may first begin to cough, and then experience dyspnea and increased body temperature which can lead to hypoxemia with a decrease in oxygen saturation and ultimately to acute respiratory failure syndrome.

A diagnosis of PB is usually established after expectoration of bronchial casts [5]. Bronchoscopy is the most important diagnostic and therapeutic method; both computed tomography and magnetic resonance lymphangiogram images might be helpful [18,19].

The aim of this study is to present a retrospective analysis and follow-up (e-mail and/or telephone interviewing) of patients who had been admitted to the Institute of Tuberculosis and Lung Disorders in 2005–2019 due to PB. The intention of the authors was to draw attention to the fact that this rare complication may appear not only in association with the Fontan palliative surgical procedure but also in patients without CHD.

## 2. Materials and Methods

### 2.1. Patient Selection

Between 2005 and 2019 at the Institute of Tuberculosis and Lung Disorders (Rabka, Poland) 14 cases diagnosed with PB were observed, i.e., 10 patients with associated CHD (Group A, 6 females and 4 males) after the Fontan procedure and 4 patients without heart defect or cardiosurgical intervention (Group B, 3 males and 1 female) (Table 1). In Group B (four cases), 2 children had been diagnosed with asthma, while 3 patients had a severe lower respiratory tract infection in their history. Patient demographics and information about first presentation of PB episode and diagnosis were obtained by reviewing hospital charts (Table 1). Since our study is a retrospective review, the institutional review board waived the need for an IRB protocol.

### 2.2. Histopathological Examination

Cast sample collection and processing for pathological assessment and histologic section staining with hematoxylin and eosin were conducted, as described in [20]. The cast sections were stained for fibrin using a Martius Scarlet Blue kit (BioGnost, Zagreb, Croatia). The stained sections were inspected by skilled and experienced hospital pathologist.

### 2.3. Immunological Assays

Lymphocyte subsets in peripheral blood were assayed, as described in [21]. IgG1 subclass determination was performed by enzyme-linked immunosorbent assay using a Human IgG1 ELISA Kit (Abcam, Cambridge, UK).

### 2.4. Intermediate-Term Follow-Up

The parents or guardians of patients were contacted by e-mail or telephone between September 2017 and June 2019 to enquiry about recurrence of PB and strategy of treatment. The questions asked during follow-up period after the last PB episode are provided in Table 2.

### 2.5. Statistical Methods

Descriptive statistics were used to sum up demographic, historical variables and data related to the clinical manifestation of PB, cast location, as well as their microscopic pathology, management, and treatment. A Student’s *t*-test was used to compare the number of PB episodes in both groups and age of developing PB symptoms.

Clinical characteristics of patients with CHD and PB (Group A) were compared with PB in children without heart defect or cardiosurgical intervention (Group B), and the results are shown in Table 1 and Table 3.

## 3. Results and Follow-Up

### 3.1. Clinical Features at Presentation

In Group A, the PB-affected patients were younger; their age ranged from 2.1 to 10.9 years (5.6 on average vs. 9.43 in Group B, *p* = 0.1160) (Table 1). The period between the Fontan procedure and the development of this complication for the first time ranged from 0.4 to 5.3 years (1.9 year on average, data not shown). All children had a cough. The expectoration of huge amounts of thick sputum and the formation of bronchial casts was observed in all patients in Groups A and B (Table 3), whereas dyspnea occurred in 50% of the investigated children (Group A, 5/10 and Group B, 2/4, data not shown). The example of bronchial cast expectorated by patient #4 is shown in Figure 1.

### 3.2. Diagnostic Investigation

All patients underwent a bronchoscopy. The diagnosis was based on the expectoration (spontaneous or during bronchoscopy) of endobronchial plugs. In Group A (Table 3), the localization of bronchial casts on the right side only was found in 60% of patients (6/10), on the left side only in 20% (2/10) of patients, while bilateral cast and that in trachea were observed with a frequency of 10% each (1/10 and 1/10, respectively). In Group B (without underlying cardiac disease), 100% patients (4/4) had bronchial casts only on the left side, and none on the right side (0%) nor bilateral (0%). A half (50%) of these patients experienced the involvement of left superior lobar bronchus (2/4), i.e., 25%, left inferior lobar bronchus (1/4) and 25% suffered from involvement of both left superior and inferior lobar bronchi (1/4). No patient in Group B had any cast on the right side (Table 3).

The expectorated casts were transferred to histopathological examination. We observed mainly fibrin-containing casts (80% in Group A vs. 50% in Group B), while inflammatory cells were present in all patients (Table 3). Bacteriological examinations revealed positive sputum culture in 40% (4/10) of patients in Group A, while in Group B, the positive culture was obtained in 50% (2/4) of children. In those patients who had a positive sputum culture, the most commonly detected microorganism in both groups appeared to be *Haemophilus influenzae* (3/10 in Group A and 1/4 in Group B). Children for whom data on the use of antibiotics were available represented 50% of total patients in Group A and 75% in Group B (Table 4).

Immune dysfunction was diagnosed in 30% of the children in Group A (Table 1). It took the form of an aberrant lymphocyte population: a reduction in T CD4/CD8 lymphocyte count (Case 1), a reduction in the number of T CD4/CD8 lymphocytes and deficiency of IgG1 (Case 2) or anatomic asplenia (Case 9). Immune deficiency accompanied by a decrease in CD4/CD8 lymphocyte count was also diagnosed in 25% of patients (Case 1) in Group B (Table 1).

### 3.3. Medical Therapy

During hospitalization in the Department, the physiotherapy of the respiratory system was essential; however, various inhalations were also performed. In Group A, inhaled medications for acute PB exacerbations involved mucolytics and corticosteroids in four patients (4/10) each, hypertonic saline and β_2_ adrenergic receptor agonists in two patients (2/10) each (Table 4). Six patients received inhaled heparin (Cases 2, 4, and 7–10; in Case 4, the treatment was continued for the period of 3 months after hospital discharge) and two patients were treated with dornase alfa (Cases 4 and 5). The chronic pharmacotherapy included hypertonic saline inhalation (4/10) and corticosteroids (4/10). The β_2_ adrenergic receptor agonists were administered in the case of emergency (Table 4).

In Group B, the acute and chronic management consisted of both mucolytics (4/4) and corticosteroid inhalations (4/4). Additionally, salbutamol was administered in the patient #13 (Table 4). No patient in Group B received inhaled heparin nor hypertonic saline at any time.

### 3.4. Long-Term Outcomes

PB had a recurrent character in the majority of patients in both groups (10/14) (Table 4). A single event of PB was observed in three patients in Group A (cases 5, 6, 10), and in one patient in Group B (case 12). Despite chronic therapy, seven out of ten patients with CHD (Group A) had a relapse (within 4 months to 5 years). In Group B, the recurrence of the disease was observed in three out of four patients within 1 to 10.8 months. On average, time to recurrence in Group A was 1.47 vs. 0.265 years in Group B.

Due to insufficient therapeutic progress during hospitalization, inhaled heparin was applied in 60% of patients in Group A (Table 4), which was then continued for three months, and afterwards only when symptoms appeared. In Case 4, the rebound of PB was not reported for 2.5 years, and in Case 8 only for 8 months. However, in another patient (Case 2) the improvement was not observed and both cough and dyspnea, with the expectoration of fibrinous plugs, occurred almost every month.

## 4. Discussion

Our research presents 14 patients with PB, including ten cases after the Fontan procedure. We observed that the first episode of PB in children with CHD tended to occur at a relatively younger age than in the non-CHD patients. In four patients without CHD, the bronchial casts were located only on the left side.

### 4.1. Age and Presumed Immune Deficiency at First Presentation of PB

In previous studies [4,19,22,23], PB in patients after the Fontan procedure was recognized for the first time, on average, at between 3 and 7 years of age, while for the remaining cases the age ranged from 4.9 to 12.8 years. A similar pattern was observed in our cohort (Table 1). One of the reasons why the children with CHD tended to be younger might be an excessive leakage and retention of lymph in the bronchi related to preexisting lymphatic abnormalities [24,25] exacerbating the circulatory complications after cardiosurgical intervention [3,4]. The increased diameter in lymphatic vessels at lung biopsies [6] as well as in the images of dynamic contrast-enhanced magnetic resonance lymphangiography [26] seem to indicate that lymphatic disorders may be involved in the pathogenesis of PB.

Other factors which may likely contribute to cast development seem to be related to immune dysfunctions which were diagnosed in 30% of the patients in Group A and also in 25% of the children in Group B (case 11). A half of the children in Group A (50%) and the majority of the children (75%) in Group B were also suffering from respiratory infections that were treated with antibiotics (Table 4). Positive sputum culture was detected in 40% of the children in Group A and 50% of the children in Group B. In addition, the inflammatory cells were the main ingredients in the airway casts in all children. No patients in Group A had atopy/allergy/asthma, while all children in Group B presented a chronic coexisting inflammatory illness such as asthma or allergy (Table 1).

It seems likely that inadequate resolution of inflammation [27] manifested by the low number of T cells [28] and the presence of proinflammatory cytokines in plastic bronchitis casts [27] may exacerbate pulmonary lymphatic leakage, leading to the retention of content in the bronchial lumen. The activation of human lung mast cells may induce the release of lymphangiogenic factors including vascular endothelial growth factor C [29] which can result in pulmonary lymphangiectasia and pleural effusions [30]. The structural abnormalities of lymphangitic vessels with lymphatic endothelial cells lacking discontinuous intercellular junctions normally found in initial lymphatics, which were necessary for efficient fluid entry, was detected in the initial lymphatics at sites of inflammation [31]. In addition, mutations or functional defects affecting the lymphatic vessels [32,33] may further predispose to impaired pulmonary lymphatic circulation.

### 4.2. Lymphatic Disturbances in PB

Despite the fact that the first author of PB’s scientific description was Claudius Galenus (130–200 A.D.) [22], the pathogenesis is still unknown and the pathomechanism may be complex. Brogan et al. [23] proposed the division of patients with PB into three groups, depending on basic disease, i.e., CHD, asthma, or allergy, and finally those without these medical burdens. One hypothesis assumes that an increase in the venous pressure has considerable impact on lymph circulation disorders resulting in significant distension of vessel (lymphangiectasia) or their excessive proliferation (lymphangiomatosis) [34]. On the one hand, lymphangiectasia is a congenital pathology, which frequently occurs in some clinical syndromes (i.e., Down, Noonan, and Turner syndrome). On the other hand, it may be secondary to cardiovascular diseases or, occasionally, viral infections [4]. Thoracic duct overflow is suspected of causing excessive protein-rich plasma fluid leak, accompanied by increased permeability of respiratory tract mucosa. Similar phenomena in the intestines might result in protein losing enteropathy (PLE) [35]. Intestinal lymphangiectasia is perceived as an initiating factor of PLE in children after the Fontan procedure [4]. We observed fibrin residues in 80% of the children in Group A, which appeared to confirm the role of lympho stasis in cast formation after Fontan surgery. The presence of fibrin in the bronchial casts in half of the non-CHD patients with PB (Group B, Cases 12 and 14) (Table 3) may likely reflect a local inflammation-dependent lymphatic transport dysfunction [36].

PB and chylothorax are the most frequent clinical complications of pulmonary lymphatic perfusion syndrome (PLPS). The complex of these disorders leads to an improper lymphatic flow through the lymphatic vessels of a respiratory system. Imaging studies such as lymphoscintigraphy, dynamic contrast MR lymphangiography (DCMRL), or conventional lymphangiography have confirmed the relationship between lymphatic disorders and PB in patients after Fontan procedure [25,26,34]. One of the treatments of PB include percutaneous lymphatic interventions such as embolization of the thoracic duct [6,34], a surgical intervention for reducing the central venous pressure, and pharmacological management. A diet based on minimizing long-chain triglycerides intake and supplementation of medium chain triglycerides (MCT) associated with lymphatic interventions may enable a reduction in a cast production or other complication of Fontan procedure such as thromboembolism [6,34].

### 4.3. Role of Immunological Factors in PB

The frequent presence of thromboembolism in Fontan circulation seems to form casts in the respiratory tract via secretion of inflammatory mediators [4,35,37,38]. Genetic factors are also considered to be involved in a chronic inflammatory state [4]. This aspect may be especially important in patients without CHD; however, the family cases of PB have not been described yet.

Inflammatory cells were found in every histopathological examination of children in both groups; however, fibrin was the core component in Group A. Madsen et al. [4] proposed a classification scheme based, firstly, on comorbidities and, secondly, on cast histology if the associated disease remained unclear. The cast analysis in patients after Fontan surgery, including our results, evidenced both a high predominance of lymphocytes and fibrin. A widely used classification system includes underlying etiology of PB: primary lymphatic disorders, non-lymphatic disorders, or structural CHD [4,20,39]. 

Recently, Liptzin et al. [40] suggested a classification scheme based on the presence (Class II) or absence (Class I) of fibrin, which appeared to be helpful in treatment guiding. In the case of Class I PB, where there is an absence of fibrin, treatment should focus on mucolytics and other accessory measures based on cast pathology for bacteria and inflammatory cells. For Class II PB, where fibrin is present in the casts, pulmonary fibrinolysis with pulmonary anticoagulation has been recommended in addition to usual supportive care. Although we did not use fibrinolytic drugs to disintegrate the formed casts, 60% (6/10) of our patients in Group A were inhaled with the constant dose of heparin (5000 IU, independent on child weight) (Table 4) to prevent or lessen the plastic cast formation. The patients in Group II who did not receive heparin were mainly subjected to mucolytic and corticosteroid medication (Table 4).

### 4.4. Medical Intervention in Children with PB

All patients underwent the bronchoscopy, which was considered necessary for diagnostic and therapeutic purposes during acute exacerbations of PB [4,19]. Sometimes a dense mucofibrous formation occluding the pulmonary bronchial tree could not be removed by flexible bronchoscopy (Cases 5, 7, and 8). Rigid bronchoscopy with foreign body forceps and a suction was required. Similarly, in Case 2 (which had been initially described by Lis et al. [1]), the thick bronchial casts were also removed by the rigid bronchoscopy. Ata subsequent episode of PB in this patient, the plugs were partially removed by two flexible bronchoscopic procedures, but the airways were not completely cleared until inhalation of dornase alfa was administered.

Soyer et al. [41] presented a retrospective analysis of five patients with PB, of whom none had underlying CHD. The redundant bronchial casts had led to dyspnea despite intensification of chronic pharmacological treatment [41]. All patients underwent serial rigid bronchoscopic interventions, sometimes with additional tools such as forceps, because of a partial or complete airway obstruction. The serial rigid bronchoscopy was evaluated as a safe and effective method in the patients who become unresponsive to the standard treatment. Interestingly, the most common localization of the cast formation was the left main stem bronchus [41]. Similarly, in our study, among all patients without CHD, the plugs were also observed only on the left side. This phenomenon can be explained by the fact that the thoracic duct usually drains at the posterior aspect of the left internal jugular and subclavian vein confluence [3]. Although we did not look for lymphatic disorders in our cohort, we anticipate that an impediment in lymphatic flow in the rare cases of a dilation/collateralization of lymphatics may lead to lymph seeping to the surrounding lung tissue which could stimulate enhanced cast development in bronchial tree, likely also in non-cardiac patients. This process can be severely exacerbated by a coexisting respiratory insufficiency or inflammation, as seen in our non-CHD children. A prevalence of bronchial cast on the right side in Group A (CHD patients) may result from possible disruption of lymphatics at the time of surgery because the Fontan conduit is usually implanted on the right side.

A placebo-controlled trial on acute exacerbation and long-term prevention of cast formation has not been conducted yet because of a relatively low PB prevalence. Respiratory physiotherapy was an integral part of the therapy. Mucolytic drugs (N-acetyl cysteine, ambroxol, and hypertonic saline), corticosteroids, auxiliary bronchodilators, and antibiotics were used to facilitate bronchial cast disruption and evacuation. Three patients (Cases 4, 5, and 11) were unresponsive to this standard treatment, and therefore inhaled dornase alfa (typically used in cystic fibrosis therapy) was administered (Table 4). There is no evidence for efficacy of this agent in PB, and therefore using dornase alfa should be considered as an empirical trial.

Inhaled heparin during hospitalization was applied in 60% of the patients in Group A and in no patients in Group B (Table 4). In rare cases (Case 1), we observed a lack of satisfactory results, likely due to a high level of fibrin in the bronchial casts. Heparin does not have a fibrinolytic effect, but instead shows anti-coagulation properties, preventing fibrinous cast formation. Therefore, this therapy is suitable for patients with the fibrous bronchial casts.

Recurrences of PB remain a serious problem. Despite the pharmacological management of chronic PB, they were observed in 70% (7/10) of the cases in Group A and 75% (3/4) of the cases in Group B (Table 4). Corticosteroids and mucolytics were frequently applied to prevent disease recurrence. A relatively short time to recurrence in Group B may indicate indirectly that the chronic process still continues. Lis et al. [1] proposed the use of hypertonic saline (3%) which, due to cast moisturizing, facilitated the cast expectoration by the patient and prevented recurrence of PB for up two years.

In our population, deaths were not observed, however, the observational time was relatively short (2.5 months–8 years). According to the relevant published literature, PB is characterized by a high rate of mortality, i.e., about 50% [1]. The available data are inconclusive; however, the worst prognoses have been observed in patients after the Fontan procedure [42]. Brogan et al. [23] conducted the follow-up of their 42 patients with PB over a much longer period of time (1991–2001). Their patients were divided into three groups based on their underlying illness: thirteen patients had asthma or allergic histories, seventeen patients had cardiologic abnormalities, and twelve patients who developed PB had neither asthma nor cardiac disease. The mortality rate turned out to be considerably higher in children with CHD than in the remaining groups [23], which could be explained by the coexistence of respiratory and circulatory disorders. It has been reported [43] that asthma is a risk factor for recurrent PB in non-CHD children.

The limitation of our study is its retrospective character and a limited number of patients. Another limitation could be a lack of influenza infection evaluation, as suggested by some reports [11].

## 5. Conclusions

Overall, PB is a rare condition with poorly understood etiology. There are still uncertainties regarding the pathomechanism, and hence the means of appropriate treatment. It should be pointed out that PB also occurs in patients without cardiac problems. There are some differences in the course of the disease in non-CHD patients, including more advanced age of the first PB episode, and a likely stronger role of inflammatory factors and mechanisms. We should consider this disease during a differential diagnosis, especially if a dry cough with asymmetry of auscultatory changes over the lung fields is observed. This would allow us to recognize the illness early and institute the appropriate therapy right away, which may help to improve the prognosis. Patients with a past medical history of PB require a periodical medical assessment and immediate attention if a dyspnea or bronchial casts occur and/or saturation of mixed venous blood decreases. In this regard, the magnetic resonance lymphangiography is an appealing mode for the assessment of likely lymphatic disturbances, also in non-CHD children with plastic bronchitis. It seems that creating a registry of plastic bronchitis cases could contribute to the development of more rapid diagnoses and effective treatment.

## Figures and Tables

**Figure 1 jcm-11-00044-f001:**
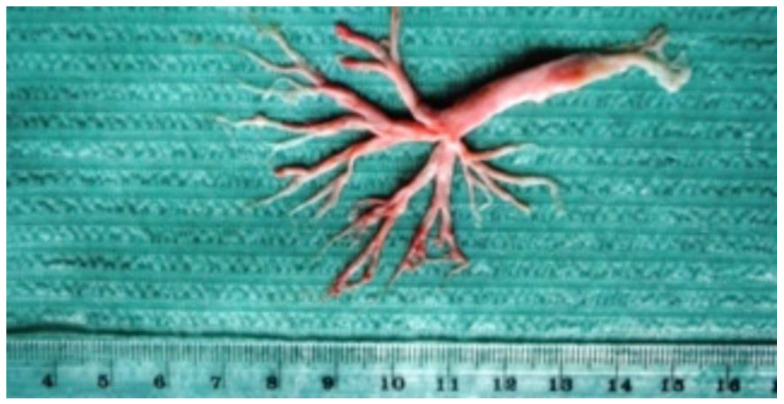
A bronchial cast expectorated by patient #4 with HLHS (hypoplastic left heart syndrome).

**Table 1 jcm-11-00044-t001:** Patient demographics.

Patient Identification	Age at First Presentation (year)	Sex	Underlying Disease Process	Immunological Deficiency
Group A				
Case 1	10,9	M	CAVC defect, PS (heterotaxy syndrome), left hemiparesis	Derangement in lymphocyte proportion
Case 2	3.6	F	DORV, TGA	Hypogammaglobulinemia, deficiency of IgG1 subclass, decrease in CD4/CD8 lymphocytes
Case 3	8.8	M	HLHS, paralyses of phrenic nerve	None
Case 4	4.2	F	HLHS, left vocal fold paresis	None
Case 5	4.1	F	VSD/TV	None
Case 6	3.8	M	PA/VSD/ASD II/TGA	None
Case 7	4.8	M	PA, DILV	None
Case 8	9.4	F	HLHS	None
Case 9	4.3	F	HLHS, TGA, AVC, PS, TAPVC, dextrocardia	Anatomic asplenia
Case 10	2.1	F	HLHS, TI	None
Average	5.6			
SD	2.96			
Group B				
Case 11	4.8	M	Inhaled allergy (cat), asthma, allergic rhinitis, atopic dermatitis	Decrease in CD4/CD8 lymphocytes
Case 12	17.5	F	Inhaled allergy (house dust mites), asthma, allergic rhinitis, severe recurrent pneumonia in the past	None
Case 13	6.4	M	Asthma, adenoid hypertrophy, severe respiratory failure in the past	None
Case 14	9.0	M	Severe respiratory failure in the past, asthma suspicion	None
Average	9.43			
SD	5.65			
*p* (*t*-test, unpaired)	0.1160			

Abbreviations: AVC, atrioventricular canal; CAVC, complete atrioventricular canal; DILV, double inlet left ventricle; DORV, double outlet right ventricle; HLHS, hypoplastic left heart syndrome; PA/VSD/ASD, pulmonary atresia/ventricular septal defect/atrial septal defect; PS, pulmonary stenosis; TAPVR, total anomalous pulmonary venous return; TGA, transposition of great arteries; TI, tricuspid incompetence; VSD/TV, ventricular septal defect/tricuspid valve.

**Table 2 jcm-11-00044-t002:** Questions asked by interview or e-mail with parents/guardians during the follow-up period after the last plastic bronchitis (PB) episode.

1. Did your child develop another PB episodes (after hospital discharge)? If yes—what was the period of time between episodes? How was it managed?
2. Has your child used/uses chronic therapy? How long? What was/is the therapy? Was the therapy modified?

**Table 3 jcm-11-00044-t003:** Airway cast location and histopathologic characterization.

Case	Cast Localization	Microscopic Pathology	Sputum Smear Examination
Group A			
Case 1	Trachea	Fibrin residues, neutrophils	*Haemophilus influenzae*
Case 2	Left superior lobar bronchus	Fibrin residues, lymphocytes	Negative result
Case 3	Right superior lobar bronchus	Fibrin residues, neutrophils, macrophages	Negative result
Case 4	Right middle lobar bronchus	Fibrin residues, lymphocytes	*Haemophilus influenzae*
Case 5	Right superior lobar bronchus	Fibrin residues, lymphocytes	Negative result
Case 6	Left main stem bronchus	Fibrin residues, neutrophils	*Moraxella catarrhalis*
Case 7	Right inferior lobar bronchus	Fibrin residues, macrophages	Negative result
Case 8	Right middle lobar bronchus Right inferior lobar bronchus	Fibrin residues, macrophages	Negative result
Case 9	Right middle lobar bronchus Right inferior lobar bronchus	Macrophages, neutrophils, epithelial cells	Negative results
Case 10	Left superior lobar bronchus Right superior lobar bronchus	Macrophages, neutrophils, epithelial cells	*Haemophilus influenzae*
Group B			
Case 11	Left superior lobar bronchus Left inferior lobar bronchus	Neutrophils	Negative result
Case 12	Left superior lobar bronchus	Fibrin residues, lymphocytes, granulocytes, macrophages	Negative result
Case 13	Left inferior lobar bronchus	Neutrophils	*Pseudomonas aeruginosa*
Case 14	Left superior lobar bronchus	Fibrin residues, lymphocytes, granulocytes, macrophages, eosinophils	*Haemophilus influenzae*, *Staphylococcus aureus*

**Table 4 jcm-11-00044-t004:** Recurrence characteristics in patients with plastic bronchitis.

Case	Number of Episodes	Time to Recurrence after First Episode of PB (years)	Antibiotic Therapy for Acute PB Exacerbation *	Medical Therapy for Acute PB Exacerbation ^†^	Chronic Inhaled Outpatient Therapy ^†^
Group A					
Case 1	2	1.3	Amox/Clav	Amb (inh)	Amb (periodically), Cort, PT
Case 2	3	5	Unknown	Amb (inh), Hep (inh)	3% HTS, F/I, Hep (from second episode)
Case 3	2	4	Unknown	Bud (inh)	Bud, Slb (as needed), PT
Case 4	3	1.5	Unknown	Hep (inh), Slb (inh), DA (inh), NAC (po)	DA, Slb (as needed), 3–7% HTS, Hep
Case 5	1	0	Amox/Clav, Azm	3% HTS (inh), Amb (inh), Bud (inh), Slb (inh)	Slb (as needed), DA
Case 6	1	0	Amox/Clav	3% HTS (inh)	3% HTS, Bud (periodically)
Case 7	3	1	Unknown	Bud (inh), Hep (inh)	Bud, Slb (as needed), Hep (periodically)
Case 8	2	1.6	Unknown	Hep (inh), Bud (inh), F/I (inh), NAC (po)	Hep (periodically), Amb (periodically)
Case 9	5	0.3	CFX	Hep (inh), F/I (inh), Amb (po)	3% HTS (periodically), Hep (periodically)
Case 10	1	0	Amox/Clav	Hep (inh), F/I (inh), NAC (po)	Hep
Average	2.3	1.47			
SD	1.25	1.73			
Group B					
Case 11	2	0.08	Caz, Clr	Amb (inh), Bud (inh), DA (inh)	Bud, Amb (for 3 months)
Case 12	1	0	Unknown	Amb (inh), Bud (inh)	Amb, Bud
Case 13	2	0.9	Cxm, Azm, Gen (inhaled)	Amb (inh), Bud (inh), Slb (inh)	Amb, Bud
Case 14	8	0.08	Cxm, Amk, Tmp-Sxt, Amox/Clav	Amb (inh), Bud (inh)	Amb, Bud
Average	3.25	0.265			
SD	3.20	0.425			
*p* (*t*-test, unpaired)	0.4224	0.2035			

* Antibiotic therapy administered during acute exacerbation during the first PB episode. ^†^ Abbreviations:Amb, ambroxol; Amk, amikacin; Amox/Clav, amoxicillin with clavulanic acid; Azm, azithromycin; Bud, budesonide; Caz, ceftazidime; CFX, ceftriaxone; Clr, clarithromycin; Cort, corticosteroid; Cxm, cefuroxime; DA, dornase alfa; F/I, ipratropium bromide+fenoterol; Gen, gentamycin; Hep, heparin (5000 IU); HTS, hypertonic saline; inh, inhaled; NAC, N-acetylcysteine; po, per os; PT, physiotherapy; Slb, salbutamol; Tmp-Sxt, trimethoprim-sulfamethoxazole.

## Data Availability

The original data presented and discussed in this article are contained in the body of the article. The data presented in this study are available on request from Henryk Mazurek.

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
