# Peer review of "Plastic Bronchitis—A Serious Rare Complication Affecting Children Only after Fontan Procedure?"

_jcm, 2021, doi:10.3390/jcm11010044_

Round 1

Reviewer 1 Report

This retrospective report of 14 cases of cardiac and non-cardiac PB describes treatment modalities and clinical course and reminds the readership of the non-cardiac PB patient group’s existence. Whilst this report does not strictly add anything new to the body of knowledge on the subject per se, case series of rare conditions are nevertheless welcome publications. The ‘review of the literature’ component of the report has significant scope for improvement.

The entire report needs a proof read by a native speaker for English language, particular attention should be paid to the discussion, which is difficult to follow in places. There are minor but multiple phrasing and grammatical corrections needed. There are a few typos.

The number of patients are too few and heterogeneous for any meaningful statistical analysis, though the methods and statstics used are appropriate.

The introduction could be more concise but contains relevant content and references to contextualise the report.

Results are reported logically, tables are clear and relevant.

Both the results and discussion would benefit from structure, possibly in the form of subheadings eg: Clinical features at presentation/Diagnostic Investigation/Acute Medical therapy/Chronic medical therapy/ Long term outcomes.

Overall, the discussion lacks structure, jumping around from discussion point to discussion point eventually spilling into a summary of sorts but no clear conclusion.

The authors do not represent a clear grasp of the current perspectives on PB in the post-Fontan patient - whilst the basic medical therapy of PB is established, (albeit inconsistently from unit to unit), the treatment modalities for recurrent persistent PB (both old and new) are squarely aimed at decompressing venous and lymphatic hypertension;  such as Sildenafil, Fontan fenestration (Chaudhary 2004 Heart), Innominate vein turndown to decompress the thoracic duct (Hraska 2020 and 2021 semin thorac cardiovasc surg paediatr), Fontan takedown, or heart transplant (Schumacher JAHA Risk Factors and outcome of Fontan associated Plastic bronchitis: a  case control study 2014).  

Have any of the patients with recurrent PB been transplanted or referred for transplant? It is fairly unusual for all Fontan associated PLE’s to still be alive when 5 year mortality/transplant rates are reported to be around 50%, are the authors 100% sure there aren’t any patients missing from the cohort?

There is some consideration that those with PB after Fontan are more likely to have developed arterial collateralisation to the lungs, it is unclear if this is a cause or an association but CMR often shows accompanying lymphatic abnormalities. There is an emerging trend to performing T2 weighted CMR, dynamic lymphangiogram for diagnostic purposes in Fontan PB and PLE now, and various centres globally are attempting to establish lymphatic intervention programs, this should be reflected somewhere in the summary and closing statements.

The lateralisation of casts to the left in non-cardiac patients is interesting, CHD patients often cast from the right, we have often assumed this is related to the inflammation/possible disruption of lymphatics at the time of surgery – the Fontan conduit is typically (not always) implanted on the right (to the IVC and RPA), similarly the second stage surgery (Glenn shunt) is usually an operation carried out in the right thorax, and those that show prolonged pleural drainage or chylothorax often do so predominantly from the right side suggesting disordered lymphatics are underway from as early as the first few months of life. There is work to show that the lymphatics of the fetus with HLHS is abnormal, with most severe cases having lymphangectasia at birth, it is likely there are antenatal contributors that we do not understand yet. (See: Saul 2016 Pediatr Radiol HLHS and the nutmeg lung pattern in utero). It would be good for the authors to dig around for a hypothesis on why the non-cardiac tend to cast from the left, perhaps the different angle of the left bronchial tree makes for more frequent mucus plugging and lymphatic irritation? Are there other reports of non-CHD PB that describe the same laterality phenomenon? Is there anything intrinsically different about the lymphatics of the left lung? Have any of these patients had their thoracic duct anatomy (left sided – usually) imaged/investigated? TD patency confirmed?

169: Dynamic lymphangiography and IR lymphatic interventions are an important emerging treatment modality in CHD PB, with good papers available to reference, you mention this but only fleetingly and with no references ( Ref number 5 is important, Yoav Dori group have other important publications on the subject eg MRI of lymphatic abnormalities after functional single ventricle palliation surgery 2014 ARJ)

Line 132: This comment about mucus secretion after cardiac surgery very much misses the point. The Fontan operation is usually performed around the age of 3 and the onset of PB is often around a year or so after the operation, this is the major determinant in age of onset. It is not broadly accepted that mucus production is the key component of PB, in CHD, and it is not a major feature a year down the line after Fontan surgery. What is a feature, is abnormally developed and (probably) hypertensive lymphatics (try: Ghosh et al 2020 JAHA Prevalence and cause of early fontan complications: Does the lymphatic circulation play a role?). Rather than increased mucus production in the lung, we think that when the fontan is done the IVC territory experiences chronic venous hypertension, this in turn causes lymphatic hypertension, increased lymph production and worsening lymphatic stasis/congestion, this lymph then has to drain via the TD which connects to the SVC which will have a higher pressure post fontan than Pre-Fontan, exacerbating the dysfunctional lymphatics. (In some, and we don’t understand who yet), the result is a lymph&protein leak, either into the gut or the lungs, accompanied/worsened by by a poorly understood chronic inflammatory component, resulting in PB or PLE (or both). The CMR papers on the subject do show that pre-Fontan, severely abnormal lymphatics are strong predictors of fontan failure and lyphatic manifesations such as prolonged post operative pleural drainage/ chylothorax/ need for Fontan takedown/ PLE and PB.

141-144: There is a male preponderance of HLHS and all types of CHD, but the statement about airways being more narrow in boys is factually wrong, the difference changes according to alveolar volume to airway cross sectional area and indexes to body surface area differently according to age and gender. (read: Sex differences in respiratory function LoMauro/Aliverti in Breathe, and Sheel 2016 Revisting Dysanapsis: sex based differences in airways and the mechanics of breathing during exercise Exp Physiol) Also, the dysynapsis paper you reference is a lung function test only paper from 1995… more up to date options are more relevant where cross sectional imaging and spirometry investigate this topic.

Authors mention that the survival is worst in the CHD group – this is hardly surprising when we consider the hypotheses around the root cause of cast formation, in the respiratory group inflammatory processes are thought to be the key player – largely treatable/known treatment modalities. In the CHD group abnormal lymphatics, chronic venous hypertension, arterial collateralisation, possible antenatal lymphatic abnormalities and possible/likely contribution from abnormal diastolic cardiac function causing pulmonary venous hypertension all are hard to treat/no established agreed approach (apart from cardiac transplant which still often has poor outcomes by the time it is carried out owing to poor pre-operative condition). Also PB and PLE are broadly considered to be manifestations of Fontan circulation physiology failure, which in and of itself has a poor prognosis attached to it.

References are mostly relevant, some are dated and more up to date, larger studies on the topic are available and could be rather referred to. There are some typos (last ref is not numbered/referred to?)

Author Response

I am attaching revised manuscript.

Reviewer 2 Report

Reviewer Comments – Report 1:

Abstract

  1. Results: Group A and B are not defined but data is presented. In Methods/Results, define that Group A is PB with CHD, and Group B is PB without CHD.    (Line 20, Abstract)
  2. Results: “chronic treatment” is stated, but what does that mean? What was used as chronic treatment? (Line 17, Abstract)
  3. Results: Recurrences of PB were observed in both groups? (Line 18, Abstract) Also, numbers here are incorrect (see below under Results and Table).
  4. Results - from data, all patients had a recurrence – so 100% in both groups. Data here is incorrect.
  5. What is remission time – time to next recurrence? Must define. (Line 20, Abstract)
  6. Instead of stating 2 cases vs 1 case, show as percent – such as 50% (2/4) of Group B, vs 10% (1/10) of Group A – as an example. Also it is customary to stick to the order of presentation.  First discuss group A, compared to B.  Each sentence thereafter should present A first and B second.
  7. “In patients without CHD, casts were on the left side” – this sentence needs to be in comparison to the other group. Here, Group B (?) is now referred to as “patients without CHD”.  Stay with same group naming, after defining what those are, and use Group B here also.  Moreover, were the casts usually on the right in patients with CHD, or everywhere?  Should tell us what happens regarding the location of casts in the other group as well here.
  8. Conclusion - this is too long, and should be just once sentence.  Moreover, all this information here is actually results, so should be moved under Results.  Conclusion is not data, it is a summary and significance of findings.  (Lines 22-28, Abstract)

Introduction

English translation will need some help in the introduction section, to be more scientific word choices, and improved punctuation.  Need more detail on diseases mentioned and why there are mentioned.   Also ARDS is mentioned, which is usually not associated with PB so this was not appropriately explained.  Did the authors mean to say Acute Respiratory Failure (ARF) resulting from airway obstruction by casts?  ARDS refers to alveolar process, but PB is an airway process.  Could the authors further explain their claim here that PB causes ARDS, with proper justification instead of speculation (case reports referenced, etc).  Or alternatively, please change from ARDS to ARF that might result from PB.

Detailed Comments:

  1. Page 1, Line 33 – PB is defined as fibrinous and/or mucous casts, not necessarily both. Please add and/or.
  2. Page 1, Line 35 – The exact procedure here is called “Fontan palliative surgical” procedure, not Fontan’s. Please correct.
  3. Page 1, Line 36 – use “such as” instead of “like”. I am not sure why PLE is brought up here – please expand and explain what PLE is if it is mentioned.  The reader will not know.
  4. Page 1, Line 40-43 – very choppy English, will need to improve word choices and punctuation. Also, does the author mean hypoxemia instead of hypoxia?  One refers to oxygen saturation decrease (hypoxemia), the other refers to tissues having low oxygen (hypoxia – indicated by lactate level usually). Also must put reference here about ARDS association with PB – this is not a known complication of PB at all.  But acute respiratory failure is!  Please see Overall Comment above.
  5. Page 1, Line 44 – what does a delayed diagnosis of PB being established mean? Does the author mean to say that often the diagnosis of PB is delayed, and then in the next sentence, state that diagnosis is usually made upon cast expectoration or bronchoscopic removal of cast material?  Those are two separate ideas that were combined here in one very difficult to understand sentence.

Materials and Methods

There is no mention of any IRB approval for this study.  Somewhere, the authors need to state if this study was approved by local institutional review board for human studies, or if the hospital’s or university’s IRB board waived the need for an IRB protocol for reasons such as this study being a retrospective review.    This must be stated.  Also, did the authors get the information from chart review about first presentation and diagnosis, and patient demographics? This also needs to be stated in the methods section.   Also must discuss under the Methods how histopathological examination was done, but whom (certified pathologist?), under what stain (H&E?), and how was fibrinous cast defined.  How were the casts processed after being collected?  Was it placed in saline, water, glutaraldehyde, formalin, etc, and then processed for histological analysis?  Need more information on this.

Detailed Comments:

  1. Page 2, Line 55-59 - Should reference the patient demographics Table 1 here under subheading “Patient Selection”.
  2. Page 2, Lines 60-63 - Please give move detail about what was included in the follow up interviews and questionnaires. Was a standard questionnaire form used for verbal interview and email communication?  What questions were asked?  How was this type of follow up standardized between patients?  Who gave the interview?  A copy of the questionnaire should be included as a table in the main manuscript or as supplemental data on an online supplement if the journal allows.
  3. Page 2, line 62 - was the inquiry from parents about symptoms of PB recurrence (as parents may not know if there is a recurrence), and what is the difference between “remission periods” and “recurrences of PB”.  Please define both of these endpoints.

Tables:

Two tables are presented.  There are a lot of misspelled words throughout the Tables.  Please check for spelling.  (Ex: Residues not resiues, and Neutrophils not Neutrophiles, antibiotic not anitibiotic, etc.)

  1. Table 1 contains many abbreviations of CHD that are not defined. Under each Table, please use the footnotes feature to write out what each abbreviation refers to – such as PA, DORV, TGA, etc – write out TGA=transposition of great arteries, DORV= ....... etc.
  2. It is rather strange to have two tables back to back, one showing Patient Identifier via Case numbers (1-10), while the other showing case number, age at first PB diagnosis, sex, and underlying disease process. It would be best to change Table 1, and have a column for Patient Identification (Group A, Case 1-10, Group B Case 11-14), another column for Age, another column for Sex, and another for Underlying Disease Process.  This should likely be Table 1, called Patient Demographics.  It would clearly present the patients included in this case series.  Next, Table 2 should be Airway Cast Location and Histopathologic Characteristics, showing the patient Case # in first column, then cast localization, micro path and sputum smear.   Table 3 should then contain PB Recurrence Characteristics in Patients with PB.
  3. Current Table 2 – it is not clear what “Case nr 1” means. What is nr?  The patients should be identified as Case 1, Case 2, etc.
  4. Current Table 2 – nothing is in the tables for Case 2, 3, 4, 7, 8, 12, 14. Please put either “none” or “unknown”.  Currently, it looks as if someone forgot to fill out the table.
  5. Table 2 – it is not clear why time of follow-up matters on this table. What data does this give?  Is this time from the first (or is it last) PB episode when email/phone interview was conducted? Please clarify what this is.
  6. Table 2 - Remission after first episode – does this column actually indicate time of Recurrence or basically first PB exacerbation?
  7. Table 2 – Antibiotic therapy for acute PB - is this what the patients were treated with for their first PB episode, or the recurrent (2nd) PB episode?  Can abbreviate the antibiotics in the table for easier read, and but in footnotes what the abbreviations mean (Ex: Amox/Clav is an acceptable abbreviation, and so on and so forth).
  8. Table 2 – Chronic inhaled therapies – Also use abbreviations, and define them in footnotes. (Ex: 3% HTS is an acceptable abbreviation for hypertonic saline). Also, this is not actually representing what the patients received at home, but based on the Results section, this is what the patients received in the hospital during their first PB attack. Is this correct?  If so, please change subtitle in this column.
  9. Table 2 values are incorrect for statistical analysis, particularly all of Group B data. Please do appropriate averages and SD. Incorrect numbers are highlighted.

Group A              

2

0.4

1.3

3

5.8

5

2

4

4

3

3.8

1.5

1

1.5

1.5

1

1.2

1.2

3

0.2

1

2

2

1.6

5

1.75

0.3

1

1.08

1.08

Ave

2.3

2.173

1.848

SD

1.251666

1.794424

1.464216

Group B

2

8

0.08

1

0.5

0.5

2

7.7

0.9

8

1

0.08

Ave

3.25

4.3

0.39

SD

3.201562

4.106093

0.393446

Unpaired t-test p-values;

            0.4223         0.1879       0.0788 

Results

In general, the results section is filled with data that will need to be shown in a graph or table format, prior to presentation.  The authors continue to do incorrect mathematics, and show data that is incorrect based on table values shown.  Calculations are wrong.  Also, it might be worth showing statistics in terms of %, with actual numbers (such as 4/10) in parenthesis after the % number (such as 40%), for easier readability.  

  1. Page 4, lines 75-79 – This discusses age of first presentation, yet this data was not shown. The authors will definitely need a separate table and column of age at first presentation to refer to here, as previously said in a Patient Demographics Table proposed above. As the author now gives us data here that was not averaged nor presented in the tables. Where did the age ranges, averages and p-values come from?  Need to show this in a table or a graph.   All this should be said in the table:   Group A – Ave age is 2.9589 +/- 5.6 (SD), and Group B is 9.425 +/- 5.654 (SD), with p = 0.1160 (t-test, unpaired).
  2. Page 4, line 80-82 - More statistics is shown without any figures or tables where one can find this data. Here, it is unclear what 8/10 patients had in group A – was it expectoration of cast at presentation?  Is that what data is shown?  Need to but this into a table or graph. Including symptoms of dyspnea at first PB presentation (which was 5/10 and 2/4 in each group).
  3. Page 4, line 87-92 – It might be useful to show how many patients had any cast on the right only (6/10), on the right and left – that is, bilateral (1/10), the left only (2/10), and trachea only (1/10). As the authors are trying to point out that Group B is 4/4 on left, and none on the right, bilateral or in the trachea. The exact lobar locations of the casts may be good to keep in the table.  Otherwise, this is very confusing to read, as all the numbers add up to 13/10 – but only have 10 patients.
  4. Page 4, line 93-94 – it would be worthwhile to mention here that Group B did not have any casts on the right side in any of the patients.   
  5. Page 4, line 96 – how are proteinaceous casts defined? This was not in the Table at all. What does this mean, and please put data in tables if one is to present it in a Results section.
  6. Page 4, lines 101-106 – again this is data presented that is not in a figure or table. Please make more figures/tables to show data about immune dysfunction if it will be discussed here. All this will also need to go into the methods section – how this was analyzed and what was analyzed.   If immune dysfunction of any kind was present in Group A but not in Group B, this is very important information and should be highlighted in this manuscript.
  7. Page 4, lines 108-115 - it would be great to see what the patients were given during their hospitalizations, and what they were sent home on for chronic outpatient therapy.  Was there anything, other than heparin in one patient for 3 months?  Same for Group B – what were they treated with acutely, and at home for chronic therapy?
  8. Page 4, line 118 - based on Table 2, all patients had a recurrence.  Where does 10/14 come from?  From Table 2, recurrence for Group A was from 0.3 – 5 yrs, average of 1.85 years.
  9. Page 4, line 120 – again, all patients had a recurrence based on Table 2 data. For Group B, the recurrence dates ranged from 0.08 – 0.9 years (which is 1 month to 10.8 months), average of 0.39 years (aka 4.7 months)  -  data here is incorrect (not 1 - 6 months).  Authors must ensure that they do appropriate mathematical calculations prior to submitting a manuscript for publication!
  10. Page 4, line 122-124 – it is unclear where this data is coming from. Based on Table 2, all patients had a recurrence.  Again, what is remission time?  Is it time to recurrence?  It should be stated that way instead, as remission word is used for cancers, not PB.   Based on above calculation from the tables, time to recurrence is 1.85 years in Group A, vs. 0.39 years in Group B. (p=0.0788)

Discussion

Overall, the discussion section is extremely long, and should be shortened by at least half.  The authors present new data in this section, which should be moved to the results section instead and explained here about how it fits with their data.  In addition, the authors speculate too extensively about the cause of PB, followed by an extensive review of causes and other treatments of PB that were not shown in the patient case series in this manuscript (lymphatic channel interventions), without a clear indication of why the authors are focusing on this intervention.   Instead, the authors should try to explain all of their findings in the discussion section, and draw on previous literature to provide support for their explanations.  Not to provide a review of PB in the discussion. That should go into the Introduction.

  1. Page 5, line 132 – authors say excessive secretion of mucus, but the authors showed that the casts were fibrinous casts. This is incongruent. Also, why would there be more mucus in CHD patients due to circulatory complications?  How is that even related?  Would like to see more discussion on this aspect, as this is not been shown before. 
  2. Page 5, lin 134 – how are immune disorders and cast formation related? The authors seem to suggest and speculate that they are. This has never been shown. Moreover, the authors speculate that earlier diagnosis of PB with CHD is due to immune dysfunction.  Will need to see a full explanation on how the authors arrived at this conclusion, with appropriate supportive references and data.   If this is just a guess, it should not be in a manuscript and will need to be removed.
  3. Line 139-142 - male vs. female statistics were not shown in the results section.  Authors should not introduce new data in the Discussion section.  If they want to discuss this here, will need to show this in the results section first.   Also, it is quite a reach to say male predilection was seen in a group of n=4.   Sample size was too small to evaluate this.  All one can say, is that 3 out of 4 patients in the non-CHD group were male.  Can’t make conclusions on this due to low power of this case series.   Similarly, not enough patients were evaluated with asthma, atopy and respiratory disease (??? What is this????), to say one way or another.  Many CHD patients also have asthma – was this evaluated in them?
  4. Line 142-144 - Need reference for males having more HLHS and asthma. Is this known?  Also, “the reason for this” -  what does this refer to -the reason for what exactly?
  5. Line 147 – add reference for Brogan et al
  6. Line 150 – add reference for lymphatic abnormalities as cause of PB
  7. Line 154 – it is not mucus that is excessive, but plasma fluid leak. Thoracic duct overflow does not cause mucus production. This has never been shown.
  8. Line 158 – this is a false statement. Fibrin residues were also found in two patients from Group B (Case 12 and 14).
  9. Line 159 – this was not shown – absence of lymphatic system disorders? Was there any data shown about lymphatic system disorders for either group A or B?  None in Results and none in the Tables were shown.
  10. Line 162 – what does “negative phenomena at lymphatic system” mean? This is unclear, please clarify this sentence.  Also, were lung biopsies performed on the case series, where lymphatic vessels were evaluated?  It is unclear why this statement is here.
  11. Line 169 – percutaneous lymphatic interventions – this is only one of the proposed therapies to treat recurrent PB. There are many treatments. Please ensure the qualifier of “one of the treatments of PB” is added here.
  12. Line 178 – more discussion of excessive mucus, while the authors clearly showed that casts are fibrinous – plasma components. Not mucus. 
  13. Line 183 – 186 - the authors are referring here to the old classification by Seear et al (1997, AJRCCM), and not Madsen et al.  Incorrect reference here and incorrect discussion of previous classification methods of PB.   Moreover, the classification is stated incorrectly.  Type I is inflammatory, fibrin and “cellular cast” per Seear.  Type II is mucus, and non-inflammatory (no fibrin ) per Seear.  And, he incorrectly stated in that manuscript that CHD patients had Type II casts (no fibrin), which the authors here clearly show is incorrect.  As it is all fibrin positive.  Madsen showed this as well.   Madsen’s classification was entire different, and underlying diagnosis-based (ex: asthma PB, SCACS PB, lymphatic abnormalities PB, etc).
  14. Line 202 – please add reference here for Soyer et al.
  15. Line 210 - the left sidedness of cast formation in non-CHD PB patients would be very interesting to discuss here.  Authors should try to explain their findings, as that is what the discussion section is supposed to contain.
  16. Line 212-219 – use of dornase alpha was not discussed in the results section, so it is unclear why it is brought up here. Discussion is to explain the results, not to show new information.
  17. Line 222 -224 – please add H1N1 influenza and COVID-19 to this list, most important viruses that cause PB nowadays.
  18. Line 220-224 – again, this is a review, which should go into the Introduction, not discussion. This paragraph does not explain the results shown.
  19. Line 225 – what dose and frequency of heparin was used? IN line 229, mucus plugs are mentioned again  - please be careful with this.  Heparin would only be used for fibrinous plugs, it is not a mucolytic! 
  20. Line 231 – heparin is not only anti-inflammatory – it is an anticoagulant, thus, preventing fibrinous cast formation. It won’t lyse a cast, but will prevent its formation, based on how it works.  This needs to be added here.  Also, if authors will focus on heparin for PB, must show data on its use in the Results section in this patient cohort presented.
  21. Line 233 – how does Sildenafil work for PB? It should be discussed if it is mentioned. Was this used on any of the patients in the patient cohort presented?
  22. Line 235 -how does tPA and urokinase work for PB? It should be discussed if it is mentioned. Was this used on any of the patients in the patient cohort presented?
  23. Line 237 - 242 – tPA is first line treatment in many hospitals now for PB. It has no adverse effects seen to date in the literature.   The reference manuscript (#11,  Li et al), says there was none of their patients with an bleeding complications.  Thus this should be removed as stated in this manuscript, as it is an incorrect statement.  The Li et al paper mentions a theoretic complication from tPA as it “may irritate the airway, may result in hemoptysis or dyspnea”, but Li et al also did not show any data on this, nor references as to this statement.  Therefore, this is not a known fact, and cannot be referenced and stated as these authors did here.  Please remove.
  24. Line 247 – authors should discuss why HTS may have worked (3%) – it improves expectoration by moisturizing of the casts, allowing easier expectoration by the patient. Also, how do the authors arrive at the conclusion that chronic inflammation causes PB in Group B (non-CHD)?  Authors should be careful about not jumping to conclusions and guessing here, but discuss their data with possible explanations – compare and contrast possible explanations only.

Author Response

I attached responses to the reviewers

Round 2

Reviewer 1 Report

No Specific amendments. From a cardiology/CHD perspective the paper is of minimal clinical value but from a broader perspective/respiratory physician as a contribution to the body of literature on PB this paper carries some value to the readership

Author Response

Thank you for your review and positive comments.

Reviewer 2 Report

Reviewer Comments:

Abstract -  

Greatly improved Abstract. Some additional minor Comments:

  1. Line 26 – Please state # of patients from total related to recurrences of PB per group: such as 3 cases out of 10 total in Group A (30%), and 1 out of 4 total in Group B (25). Individual patient Case #s should be removed from the abstract, and instead a quicker summary shown.
  2. Conclusion - improved with having two sentences now only. However, the conclusion sentence needs to be related to the results presented, and show significance of findings of this article only.  Discussing MRI here has no place, unless data on it is part of the point of the entire manuscript, which it is not.

Introduction and Materials and Methods

Greatly improved introduction and Methods sections, no further edit suggestions.

Tables:

Very nice improvements on all the Tables.  Superb job on all these tables!  Some additional and recurring comments:

  1. Table 1 – please add Average Age of first presentation under Case 10 (new line) for Group A (5.6 +/- 96 years if SD is being used), and for group B under Case 14 (new line) (9.43 +/- 5.65 years if SD is being used, with a p-value (p=0.1160) under that – as was done in Table 4 for other data statistics. Please correct the SD values as shown.
  2. Table 4 – Group B – please remove all “nr” from “Case nr 11”, to show the designation of “Case 11”, as was done for Group A. Also, please add in formatting lines between hospital and home drug use for Group B cases, same as shown for Group A.
  3. Table 4 - while Averages are now corrected (great work!), as well as the p-values, the SD values continue to be incorrect.  Using Prism Graphpad and Excel programs to calculate the SD, they are as previously stated by this reviewer. Please correct.

Group A              

2

1.3

3

5

2

4

3

1.5

1

1.5

1

1.2

3

1

2

1.6

5

0.3

1

1.08

Ave

2.3

1.848

SD

1.25

1.46

Group B

2

0.08

1

0.5

2

0.9

8

0.08

Ave

3.25

0.39

SD

3.20

0.39

Results

Improved results section, but needs further clarification and more clear written presentation of the data included in the tables. Please see additional new comments:

  1. Line 116 and line 120 are the same, which is repetitive. I would suggest removing the first sentence, and moving up the third sentence (line 120) into first sentence position, and attaching the Table 1 reference to that.
  2. Line 129 – reference Table 3 after beginning to discuss cast location. Cannot wait to do it at the end of the paragraph.
  3. Line 133 – for clarity, please state that this is Group B at the beginning of this sentence, and that 100% (4/4) patients had casts located only on the left side, 0% on the right, and 0% bilateral. In the same structure as was stated for Group A – in line with rules of scientific writing and data results presentation.  Then, the sublocations of the left lobe can also be mentioned after that.
  4. Line 129-141 - Looking at the new data in terms of Sputum cultures, it might be better to say that 4/10 (40%) patients in the Group A had a positive sputum culture, while 2/4 (50%) had a positive culture in Group B. It really does not matter that it was H.flu or Pseudomonas of M.cat for the presentation of this manuscript, just that it was positive or negative. Then, the authors could further stratify it, if wanted to focus on H.flu, to say that of those who had a positive sputum culture, the most common organism appeared to be H.flu in both groups (well, 3 in Group A and 1 in Group B).
  5. Line 141 – antibiotics treatment cannot be stated this way, as data is missing for many of the patients – missing data does not mean negative data, just missing. Therefore, once cannot say that 5/10 patients were treated with antibiotics in Group A, implying that the other 5 were not treated, since it is not known if the other 5 were also treated.  Therefore, this will have to be said as a narrative – such as: for those patients that data was available on antibiotics use in Groups A and B, which was nearly half in both groups, all were treated with antibiotics.
  6. Line 162 – please add a statement that no patients in Group B received inhaled heparin at any time, nor HTS.
  7. Line 164 - Again, based on Table 4, column 3,  all patients had a recurrence in both groups, as time to recurrence is noted with every patient. Thus it is not 10/14, but 14/14.  If indeed there was no recurrence in 4 patients, then in column 3 of Table 4, some patients would have zero in that column labelled “Time to Recurrence after First Episode of PB (years)”.  If Case 5, 6, 10 and 12 had truly only a single event of PB, then time to recurrence would be none, as they never had a recurrence.  This would also alter the Summary Average, SD and p-value in the third column of Table 4, to Ave 1.47 vs. 0.265 years, with SD of +/- 1.73 vs 0.425  (A vs B), with a p-value of 0.2035.  But which is it?  Recurrence or no recurrence for these patients?  

Discussion

This is a greatly improved discussion section - the authors did an excellent job restructuring and reworking this section.  A few comments remain:

  1. Line 193 – 194 – This statement is not supported by the results section data presented, nor the data in the Tables. What does “serious respiratory infections” mean?  Where did the number 75% of patients with serious respiratory infections come from?  From the data shown, regarding positive cultures, please see Reviewer Comment #4 and #5 under Results section above.  40 and 50% of patients in each group had a positive sputum culture.
  2. Lines 195-196 – May need to make a stronger claim here, as backed with the data presented. No patients in Group A had atopy/allergy/asthma, while all patients in Group B did. But is this because it was not asked of the Group A patients?  May need to explain this finding.
  3. Line 197-199 - authors need to explain this a bit further.  How would inadequate resolution of inflammation exacerbate pulm lymph flow disturbances?  What would the mechanism be behind that?  How did the authors arrive at this hypothesis, and how could this be further evaluated?
  4. Line 242– a new classification was recently published for PB by Liptzin et al (Peds Pulm), which likely should be mentioned here. That manuscript also reported use of alteplase for fibrinolysis in PB patients.  Was this used in any of these patients here?  If so, it should be discussed.
  5. Line 262-269 – Can the authors explain why the majority of casts in Group A were on the right side then? The theory here of Thoracic duct being on the left and causing the casts only explains Group B’s cast distribution.  Please elaborate further about what happens in the CHD patients that most often have it on the right side.
  6. Line 283 - see comment #4 above regarding mention of use of fibrinolytic drugs.  Also, what dose of heparin was used, is this known?  Were there any bleeding side effects reported with heparin use?

Author Response

Reply to the Reviewer 2, round 2

Abstract

  1. The number of patients with no recurrences in Group A and Group B was inserted. Individual patient case #s were removed from the abstract.
  2. Discussion of MRI was omitted from the conclusion section and a sentence emphasizing the findings for Group B was placed instead.

Tables

  1. In Table 1 Average Age at first presentation for Group A and Group B was added. The correct SD values were given.
  2. The designation “nr” was removed from case 11. The formatting lines between hospital and home drug use i Group B was inserted in Table 4.
  3. The SD values in Table 4 were corrected, as suggested. The averages, SD values and p-value for column Time to Recurrence after First Episode of PB (years) in Table 4 were corrected.

Results

  1. Repetitive sentence was removed and reference to Table 1 was attached in this paragraph.
  2. Reference to Table 3 was attached at the beginning of the sentence discussing cast location.
  3. As suggested, the sentence clarifying the location of casts in group B was added before the sentence about sublocation of the casts.
  4. As suggested, this paragraph was re-worded to better describe the obtained results.
  5. The sentence was altered to reflect the fact that data was available for at least half children.
  6. The sentence about lack of heparin or HTS treatment in Group B was added.
  7. The data in the column 3 in Table 4 was corrected to show no recurrence in 4 patients, and new summary average, SD and p-value were added.

Discussion

  1. This sentence was reworked, the word “serious” was omitted from the phrase “serious respiratory infection”. Although the data on antibiotics use is missing in approximately half of the children, the number of children suffering from respiratory infection comes from data in Table 4, column 4, which shows the number of children who received antibiotics in association with PB exacerbation. The number of children in both groups with positive sputum culture was also mentioned.
  2. This sentence was improved, as suggested, because there was no information on inflammatory diseases in Group A children in hospital records.
  3. In this paragraph, we provide literature data how prolonged continuous lung inflammation may worsen the function of lung lymphatics leading to lymph leakage (and ultimately bronchial cast formation in some children).
  4. We included the new classification of plastic casts here and stressed its importance in selecting appropriate treatment in patients with PB.
  5. Frequent location of plastic casts on the right side is associated with a possible damage to lymphatics during surgery.
  6. In our cohort, heparin was used at dose 5000 IU, independent on child weight. We did not recorded bleeding in association with the heparin use. This dose was given beneath Table 4 and in the text body.